# Relationships Between Positive Leadership Styles, Psychological Resilience, and Burnout: An Empirical Study Among Turkish Teachers

**DOI:** 10.3390/bs15060713

**Published:** 2025-05-22

**Authors:** Gaye Onan, Lütfi Sürücü, Mustafa Bekmezci, Alper Bahadır Dalmış, Gözde Sunman

**Affiliations:** 1Department of Tourism Management, College of Anamur Applied Technology and Management, Mersin University, Mersin 33730, Turkey; 2Department of Business Administration, Faculty of Economics, Administrative and Social Sciences, World Peace University, Nicosia 99010, Turkey; 3Department of Military Academy, Defense Studies, National Defence University, Istanbul 34149, Turkey; mbekmezci@kho.msu.edu.tr; 4Department of Management and Organization, Aeronautical Vocational School of Higher Education, University of Turkish Aeronautical Association, Ankara 06790, Turkey; abdalmis@thk.edu.tr; 5Department of Management Information Systems, Faculty of Economics and Administrative Sciences, Kapadokya University, Nevşehir 50420, Turkey; gozde.sunman@kapadokya.edu.tr

**Keywords:** servant leadership, authentic leadership, transformational leadership, psychological resilience, burnout

## Abstract

Given the rising workloads and increased risks of burnout across various industries, enhancing employee resilience and well-being has become increasingly important. This study investigates the impact of positive leadership styles—servant, authentic, and transformational leadership—on psychological resilience and burnout levels. While prior research has examined the isolated relationships between leadership, resilience, and burnout, few studies have analyzed these variables collectively within an integrated theoretical framework. To address this gap, the present study integrates insights from the Leader–Member Exchange (LMX) theory, the Conservation of Resources (COR) theory, and the Job Demands–Resources (JD-R) model to provide a comprehensive perspective. Data were collected from 387 private school teachers in Turkiye. Statistical analyses were conducted using PROCESS Macro (Model 4), assessing both direct and mediating effects. The findings reveal that all three leadership styles enhance psychological resilience and mitigate burnout. Furthermore, psychological resilience partially mediates the relationship between positive leadership styles and burnout. These results contribute to the literature by demonstrating how positive leadership can buffer against burnout through resilience, emphasizing the need for context-specific research in the field. This study also offers practical implications for managers seeking to foster supportive work environments.

## 1. Introduction

In today’s dynamic and competitive business environment, employee well-being and job performance have become critical elements for the sustainable success of organizations. In this context, the impact of leadership styles on employees’ psychological well-being and job performance is increasingly important. Leadership styles that aim to support the development, motivation, and psychological well-being of employees have positive effects on their overall well-being and success ([59]).

Psychological resilience, an important individual resource in the organizational context, is defined as the capacity of individuals to adapt to stressful and challenging situations, to recover, and even to grow stronger through these processes ([19]). Considering the stress factors and uncertainties encountered in business life, employees with high psychological resilience experience greater job satisfaction and performance because of their enhanced ability to cope with difficulties ([50]). Such individuals are more resistant to workload and organizational stressors, reducing their likelihood of experiencing burnout ([13]). It is therefore considered essential for leaders to foster a supportive organizational climate, promote employees’ personal and professional development, and provide resources to help them overcome challenges encountered in the workplace ([62]).

Burnout, one of the negative effects of organizational stress on individuals, is a syndrome resulting from prolonged job stress and excessive workload, characterized by symptoms such as emotional exhaustion, depersonalization, and a reduced sense of personal accomplishment ([61]). Beyond its negative impact on employees’ physical and mental health, burnout also leads to decreased job performance, reduced organizational commitment, and a higher intention to leave the job ([23]). Therefore, preventing and managing burnout is crucial for both individual well-being and organizational sustainability. The literature emphasizes that leaders play a critical role in reducing employee burnout ([39]). Supportive leadership styles help employees balance their workload, enhance psychological resilience, and reduce the risk of burnout ([7]). Specifically, empathetic and supportive leadership styles that consider the individual needs of employees are associated with a lower likelihood of burnout ([79]).

This study aims to examine the relationships between servant, transformational, and authentic leadership styles—collectively referred to as positive leadership styles—and employees’ psychological resilience and burnout levels. It also seeks to determine the mediating role of psychological resilience in the relationship between these leadership styles and burnout. This study explores these relationships through multiple theoretical lenses: the relationship between positive leadership styles and psychological resilience is examined within the framework of Leader–Member Exchange (LMX) theory ([30]); the relationship between positive leadership styles and burnout is analyzed within the framework of the Conservation of Resources (COR) theory ([37]); the mediating role of psychological resilience is investigated using the Job Demands–Resources (JD-R) model ([22]).

Although previous studies have explored the links between leadership styles, psychological resilience, and burnout, most have treated these variables in isolation or focused only on two-way relationships. There is a lack of comprehensive research examining how positive leadership styles (servant, authentic, and transformational) influence burnout through the mediating role of psychological resilience. Furthermore, while theories such as LMX, COR, and JD-R have been individually applied in leadership or burnout research, few studies have integrated these frameworks to gain a holistic understanding of the interplay between leadership, resilience, and burnout.

Recent research in educational settings has expanded the application of COR and JD-R models to examine how school-specific stressors—such as emotional labor, role ambiguity, and teacher-student dynamics—interact with both personal and organizational resources ([78]; [52]). Situating this study within such theoretical foundations enhances its relevance to the school context. Moreover, the teaching profession, characterized by high levels of emotional labor and stress, remains underexplored in the context of positive leadership and psychological resilience. This study addresses these gaps by adopting a multifaceted approach that combines established theories and focuses on teachers as a key sample group, thereby contributing to a deeper understanding of how leadership can mitigate burnout through resilience mechanisms.

## 2. Conceptual Framework

### 2.1. Servant Leadership

Servant leadership is an ethically grounded, human-centered leadership style in which the leader prioritizes the growth and sustainability of the organization and its members over personal gain ([53]). Unlike traditional leadership approaches that emphasize authority and control, servant leadership focuses on using power to empower others and promote their development ([41]). Key behaviors such as active listening, empathy, continuous improvement, and mindfulness characterize this style ([38]). For example, listening involves genuinely considering employees’ perspectives, while empathy enables leaders to understand and respond to the emotions of their teams. Continuous improvement reflects a sustained effort to enhance both individual and organizational capacity, and mindfulness helps leaders remain attuned to their internal states as well as the broader organizational climate. These behaviors foster a supportive environment where employees feel valued, motivated, and encouraged to grow. Servant leaders provide opportunities for learning, encourage collaboration, and cultivate a sense of belonging. As a result, employees tend to experience greater psychological safety, commitment, creativity, and adaptability—qualities that are crucial in today’s dynamic work environments ([53]; [38]).

In Turkiye, teachers often work within rigid institutional structures, where emotional demands and limited decision-making authority intensify their professional challenges. In such environments, servant leadership offers a transformative alternative by emphasizing empathy, individualized support, and professional growth. By prioritizing teachers’ emotional and developmental needs, servant leaders can help mitigate the risk of burnout, strengthen resilience, and cultivate a culture of trust and collaboration. This leadership approach is, therefore, particularly well-suited to addressing the psychosocial demands faced by educators in the Turkish school system.

### 2.2. Authentic Leadership

Authentic leadership is a leadership style in which leaders act in alignment with their core values, beliefs, and personality, and it has attracted increasing scholarly attention in recent years ([10]). Rather than relying on imitative behaviors, authentic leaders encourage employees to express their individuality and act in accordance with their own sense of authenticity ([74]). This leadership style comprises four key dimensions: self-awareness, the ability to objectively evaluate one’s own emotions and motivations; relational transparency, fostering honest and open communication with followers; internalized moral perspective, representing a commitment to fairness and integrity; balanced processing, the capacity to make unbiased decisions by considering multiple viewpoints ([75]). Together, these dimensions allow authentic leaders to build consistent, trustworthy, and value-driven relationships with employees.

Authentic leadership has been shown to enhance employee trust, motivation, and commitment while also promoting collaboration, innovation, and alignment with organizational goals ([36]). Teachers, in particular, benefit from the psychological safety fostered by authentic leadership, as it provides the transparency and emotional integrity they often seek from school administrators. In Turkiye, where hierarchical leadership structures and frequent policy shifts are common, these expectations become even more pronounced. The moral foundation and relational openness of authentic leadership can play a crucial role in rebuilding trust and restoring psychological safety in school settings. By upholding their core values and supporting teacher autonomy, authentic leaders help reduce feelings of alienation, boost morale, and foster a culture of openness and well-being—especially in contexts where bureaucratic pressure and administrative rigidity threaten teacher motivation.

### 2.3. Transformational Leadership

Transformational leadership is a leadership approach in which the leader motivates followers, builds a shared vision and mission, supports individual development, and inspires them to achieve higher levels of performance ([15]). This model seeks not only to enhance existing processes but also to reshape the organization’s core values and cultural foundations. Transformational leaders challenge the status quo, stimulate innovation, and guide their followers through a compelling vision of the future ([31]). According to [9] ([9]), transformational leadership comprises four key components: idealized influence, where leaders act as role models and build trust; inspirational motivation, through which leaders give meaning and purpose to followers’ work; intellectual stimulation, which encourages creative problem-solving; individualized consideration, where leaders support employees’ unique personal and professional needs. These dimensions reflect a leadership style that is both visionary and people-oriented, promoting innovation, adaptability, and long-term success ([17]).

Transformational leadership holds particular significance in Turkiye, where the emotionally demanding nature of teaching intersects with a rapidly changing educational landscape. Teachers frequently encounter centralized authority, rigid curricula, and limited professional development opportunities, all of which can undermine motivation and professional autonomy. In this context, transformational leaders offer critical psychological and motivational support by articulating a meaningful vision, encouraging innovation, and recognizing individual potential. This leadership style helps teachers cultivate a strong sense of professional identity and purpose, enabling them to remain engaged despite external pressures. Moreover, it encourages teachers to reframe daily stressors as opportunities for growth, thereby strengthening resilience and fostering long-term commitment to the profession.

### 2.4. Psychological Resilience

Psychological resilience refers to the capacity of individuals to adapt, recover, and emerge stronger in the face of stress, adversity, or trauma ([51]). Beyond merely coping, it involves learning from difficult experiences and using them as opportunities for growth and personal development ([6]). Resilience is widely recognized as a dynamic process shaped by the interplay between personal characteristics, social support systems, and environmental factors. Core components of resilience include self-confidence, optimism, emotional regulation, social support, problem-solving skills, flexibility, and a sense of purpose ([33]). These traits enable individuals to manage stress effectively, sustain motivation, and adapt to changing circumstances. In organizational settings, psychological resilience is increasingly acknowledged as a vital resource. Employees with high levels of resilience are more likely to maintain performance under pressure, exhibit stronger organizational commitment, and recover more rapidly from negative experiences such as burnout or job-related stress ([47]). They also tend to demonstrate greater adaptability, higher motivation, and more effective coping strategies in the face of organizational change. Empirical studies support these findings, revealing positive associations between psychological resilience and job satisfaction, motivation, and employee retention ([63]; [68]).

Within Turkiye’s educational system, the importance of psychological resilience is amplified by the intense emotional labor, administrative rigidity, and insufficient organizational support that educators routinely face. The cultural perception of teachers as ethical and intellectual role models further compound the psychological pressure they experience. Moreover, structural inflexibility and unpredictable administrative practices often create conditions in which teachers struggle to exercise professional discretion and maintain a coherent sense of agency in their roles. In such a context, psychological resilience functions not only as a personal asset but also as a crucial buffer against systemic challenges. Therefore, understanding how organizational factors—particularly leadership styles—can foster or undermine resilience is essential for developing sustainable support mechanisms for educators.

### 2.5. Burnout

Burnout is a psychological syndrome characterized by physical, emotional, and mental exhaustion resulting from prolonged occupational stress and excessive workload, particularly among professionals engaged in high levels of interpersonal interaction ([25]). It is not merely a state of fatigue; it also entails diminished motivation, reduced performance, and strained workplace relationships ([52]). According to [56] ([56]), burnout comprises three interrelated dimensions: emotional exhaustion, or the depletion of one’s emotional resources; depersonalization, which reflects a cynical or detached attitude toward work and colleagues; reduced personal accomplishment, where individuals marked by feelings of ineffectiveness and a decline in their sense of competence. Burnout not only compromises individuals’ physical and mental health but also leads to serious organizational consequences such as increased absenteeism, decreased employee commitment and diminished overall performance ([72]; [49]). As such, burnout is widely recognized as a major threat to organizational sustainability and efficiency—especially in high-demand sectors such as education.

In the Turkish educational system, teacher burnout has become an increasingly urgent issue. Teachers often face overcrowded classrooms, bureaucratic paperwork, long working hours, and top-down administrative structures, factors that collectively contribute to chronic occupational stress. The emotional labor required to manage classroom dynamics, meet curriculum demands, and navigate relationships with parents and administrators is compounded by limited autonomy and inconsistent institutional support. These conditions closely align with the core components of burnout—emotional exhaustion, depersonalization, and diminished personal accomplishment. Furthermore, teaching holds a culturally esteemed position in Turkiye, which increases the social expectations placed upon educators while not always being matched by structural support. This disconnect makes teachers particularly vulnerable to burnout. Understanding how leadership styles and psychological resilience can buffer these stressors is therefore essential for developing effective school leadership strategies, improving teacher well-being, and enhancing retention in the education sector.

## 3. Interconceptual Relations and Hypotheses

While leadership styles and psychological resilience have been studied across various sectors, their intersection in the educational field—particularly in private schools—requires closer attention. The unique working conditions of teachers in Turkiye, including bureaucratic constraints and socio-emotional demands, call for leadership approaches that foster psychological resilience and mitigate burnout.

### 3.1. Servant Leadership and Psychological Resilience

Servant leadership is defined as a leadership style in which the leader prioritizes meeting the needs of followers, supporting their development, and enhancing their well-being. This style is characterized by behaviors such as valuing employees, actively listening to them, demonstrating empathy, and promoting their personal and professional growth ([48]; [26]). Especially in environments such as schools, where emotional labor and workload are high, servant leadership can serve as a critical resource, helping employees build psychological resilience ([26]; [5]).

According to the Leader–Member Exchange (LMX) theory, leaders who invest in high-quality relationships with their followers cultivate trust, respect, and support, which can enhance employees’ ability to manage stress and uncertainty ([30]). These relational resources act as protective factors that reinforce psychological resilience, defined as the ability to recover and adapt in the face of adversity. Servant leadership, by offering emotional support, autonomy, and growth opportunities, aligns well with this framework. This is particularly important in the teaching profession, where resilience is necessary to navigate emotional challenges and avoid burnout ([78]; [58]). Based on these theoretical and empirical insights, we propose the following hypothesis:

**H1:** *There is a positive and significant relationship between servant leadership and psychological resilience*.

### 3.2. Authentic Leadership and Psychological Resilience

The authentic leadership style emphasizes the leader’s sincerity, transparency, self-awareness, and commitment to ethical values. Authentic leaders know themselves well, are aware of their strengths and weaknesses, and demonstrate honesty, openness, and trustworthiness toward others ([74]). These characteristics enable employees to trust their leaders, and this trust environment positively affects employees’ resilience levels. Research has shown that authentic leadership has a positive and significant relationship with psychological capital ([21]) and employee resilience ([55]). Within the framework of LMX theory, authentic leadership increases employees’ psychological resilience through high-quality leader–member relationships. Specifically, authentic leadership enables the leader to empathize with employees and better understand their needs through self-awareness and commitment to ethical values. This helps employees feel more positive about themselves and fosters a constructive attitude when dealing with challenges. In addition, authentic leaders’ commitment to ethical values such as justice and honesty improves employees’ ability to cope with negative situations at work and increases their psychological resilience. As a result, authentic leadership supports employees’ psychological well-being, creating a more resilient and productive workforce. Based on this, we proposed that:

**H2:** *There is a positive and significant relationship between authentic leadership and psychological resilience*.

### 3.3. Transformational Leadership and Psychological Resilience

Transformational leadership is defined by the ability to inspire, motivate, and intellectually stimulate followers while addressing their individual needs ([9]). In teaching—a profession that demands emotional labor, adaptability, and a strong sense of purpose—transformational leadership fosters trust, engagement, and psychological empowerment ([18]; [65]). By articulating an inspiring vision, recognizing individual needs, and offering consistent support, transformational leaders help employees assign deeper meaning to their work and manage stress more effectively ([8]). According to the Leader–Member Exchange (LMX) theory, high-quality relationships between leaders and followers are characterized by mutual trust, support, and open communication ([30]). These relational dynamics enhance employees’ access to social support and feedback, which in turn boosts their self-efficacy and strengthens their ability to cope with adversity. Transformational leaders, through individualized consideration and inspirational motivation, create the type of supportive exchange relationships described by LMX theory. Empirical research supports this theoretical linkage. Studies have found that transformational leadership is significantly associated with lower stress levels ([8]), increased psychological empowerment ([65]), and greater general resilience ([1]). By encouraging personal growth, enhancing coping skills, and reinforcing employees’ belief in their competence, transformational leadership plays a key role in the development of psychological resilience. Based on this theoretical and empirical foundation, we propose the following hypothesis:

**H3:** *There is a positive and significant relationship between transformational leadership and psychological resilience*.

### 3.4. Psychological Resilience and Burnout

While psychological resilience is defined as the ability of individuals to successfully adapt and recover in the face of stress, trauma, tragedy, or significant negative life events, burnout is defined as a state of physical, emotional, and mental fatigue resulting from chronic work stress ([56]). [71] ([71]) argue that individuals with higher psychological resilience cope more effectively with work stress, recover more quickly from negative events, and exhibit fewer burnout symptoms such as emotional exhaustion, depersonalization, and decreased personal accomplishment. Similarly, psychological resilience has been found to significantly reduce the level of burnout ([80]; [78]). According to COR theory, when individuals’ resources are depleted or threatened, negative outcomes such as stress and burnout are more likely to occur ([77]). Individuals with high psychological resilience are less likely to exhibit burnout symptoms, even under stressful work conditions, because they have the necessary resources to cope. Therefore, individuals with high psychological resilience tend to view workplace difficulties as challenges rather than threats, which helps them remain more engaged and motivated at work, ultimately resulting in less physical, emotional, and mental fatigue from work-related stress. Based on this, we propose the following hypothesis:

**H4:** *There is a negative and significant relationship between psychological resilience and burnout*.

### 3.5. Servant Leadership and Burnout

Servant leadership is expressed as an effective factor in reducing burnout by increasing psychological safety and trust in the leader, especially in environments with intense job demands ([5]). Indeed, servant leadership has been found to significantly reduce burnout among employees by creating a supportive and nurturing work environment ([53]; [3]; [5]). From the perspective of COR theory, servant leadership can be seen as a crucial factor that supports and protects employees’ resources. Servant leaders prioritize the needs of their employees, offer support, encourage their development, and treat them fairly. These behaviors provide employees with various types of resources. For example, servant leaders offer social support by addressing emotional needs, increase employees’ knowledge and skills through investment in their personal and professional development, and provide positive resources by encouraging their active participation in organizational processes ([26]). As a result, servant leadership, framed within COR theory, reduces burnout levels by protecting employees’ valuable resources, helping them acquire new resources, and preventing resource depletion. Based on this, we propose the following hypothesis:

**H5:** *There is a negative and significant relationship between servant leadership and burnout*.

### 3.6. Authentic Leadership and Burnout

Research shows that authentic leadership has a significant mitigating effect on employees’ burnout levels ([57]). The presence of authentic leaders strengthens employees’ crucial resources, such as social support, self-efficacy, and self-worth, making them more resilient to job stress and less likely to show signs of burnout. Moreover, by adhering to the principles of transparency and fairness, authentic leaders reduce the likelihood of employees encountering factors such as uncertainty and injustice in the work environment, which can lead to resource depletion. In the context of COR theory, authentic leadership reduces burnout levels by supporting employees’ resources, preventing losses, and helping them develop new resources. As a result, authentic leaders’ honest, transparent, and supportive attitudes help employees feel more secure at work, manage stress more effectively, and reduce burnout levels. Based on this, we propose the following hypothesis:

**H6:** *There is a negative and significant relationship between authentic leadership and burnout*.

### 3.7. Transformational Leadership and Burnout

Transformational leadership is a leadership style in which the leader triggers change and transformation by motivating followers, creating a sense of vision and mission, supporting their personal development, and encouraging them to achieve high levels of performance. Burnout is defined as a state of physical, emotional, and mental exhaustion resulting from chronic work stress, particularly among individuals in professions that require intense interpersonal interaction ([56]). [69] ([69]) suggested that transformational leadership can help reduce burnout in general. In fact, research has shown that there is a negative and significant relationship between employees’ perceptions of transformational leadership and their levels of burnout ([69]; [70]). From the perspective of COR theory, transformational leadership can be seen as an important resource provider for employees. Transformational leaders enhance the emotional, social, and psychological resources of their employees by inspiring them, motivating them, supporting their personal development, and valuing their contributions. The more resources individuals have, the more resilient they become to negative outcomes such as stress and burnout. By providing such resources, transformational leaders increase employees’ resilience to job stress and reduce the likelihood of burnout symptoms. As a result, transformational leaders help employees better manage job stress and mitigate the effects of burnout. Based on this, we propose the following hypothesis:

**H7:** *There is a negative and significant relationship between transformational leadership and burnout*.

### 3.8. Psychological Resilience Mediating Effect

The Job Demands–Resources (JD-R) model serves as a basis for explaining the mediating role of psychological resilience in the relationship between servant, authentic, and transformational leadership styles and burnout. The JD-R model posits that the work environment encompasses both ‘job demands’ (stressors such as workload, time pressure, and role ambiguity) and ‘job resources’ (factors that enhance well-being, such as support, autonomy, and feedback). This model suggests that job demands can lead to burnout, while job resources can mitigate this effect and enhance motivation ([23]). In this context, leadership styles can be considered vital job resources, not only in the general business environment but also within school organizations, where they play a pivotal role in supporting teachers’ motivation, performance, and well-being. However, the effectiveness of these job resources depends on employees’ ability to resist stress, that is, their psychological resilience. Employees with high psychological resilience can reduce the risk of burnout by utilizing the support and resources provided by their leaders more effectively.

Servant leadership enhances employees’ sense of being valued and secure by prioritizing their needs, supporting them, and encouraging their development ([53]). Based on the Job Demands–Resources (JD-R) Model, servant leadership can be considered a ‘job resource’ for employees. Research shows that servant leadership helps employees become more resilient to job stress and challenges ([5]; [3]). Employees with high psychological resilience adapt better to stress, trauma, or negative events, cope more effectively with stressful situations, and are less affected by negative emotional reactions. In this context, the supportive and development-oriented environment created by servant leaders ([5]) increases employees’ psychological resilience by strengthening their self-confidence, optimism, emotional regulation skills, social support, and problem-solving abilities. Psychological resilience acts as a buffer against burnout, reinforcing the positive effects of servant leadership. Therefore, increasing psychological resilience reduces the risk of burnout. [76] ([76]) found that employee resilience plays a mediating role in the relationship between servant leadership and work engagement. Similarly, [16] ([16]) revealed the mediating role of resilience in the effect of servant leadership on work commitment. Within the JD-R model, [73] ([73]) determined that employees who experienced high work-related social resources (servant leadership) and high personal resources (resilience, self-efficacy) had lower levels of burnout. Based on this, we propose the following hypothesis:

**H8:** *Psychological resilience mediates the relationship between servant leadership and burnout*.

Authentic leaders are individuals who possess high self-awareness, adhere to ethical values, and demonstrate honesty and trustworthiness. These leaders inspire trust, provide support, and motivate their employees, which in turn helps employees cope more effectively with job demands. Psychological resilience is defined as the capacity to successfully adapt and recover in the face of stress, trauma, or adverse events. The environment of trust and support created by authentic leadership strengthens employees’ psychological resilience ([29]; [55]). A negative relationship has been found between authentic leadership and burnout ([57]). [2] ([2]) revealed that employees’ psychological capital mediates the relationship between authentic leadership and burnout. Psychological resilience is one of the key sub-dimensions of psychological capital. [42] ([42]) found that authentic leadership has positive effects on employees’ psychological capital, which in turn reduces their job burnout. Furthermore, the same study revealed that all sub-dimensions of psychological capital partially mediate the relationship between authentic leadership and burnout. Based on this, we propose the following hypothesis:

**H9:** *Psychological resilience mediates the relationship between authentic leadership and burnout*.

Transformational leadership is a leadership style in which the leader initiates change and transformation by motivating followers, instilling a sense of vision and mission, and supporting their personal development. Transformational leaders exhibit charismatic traits, offer inspirational motivation, promote intellectual stimulation, and demonstrate individualized consideration ([9]). By providing employees with meaningful purpose, continuous support, and developmental opportunities, transformational leaders enhance employees’ capacity to cope with job demands and mitigate the negative effects of stress ([18]). Their inspiring visions, supportive behaviors, and growth-oriented approaches ([1]; [40]) strengthen employees’ psychological resilience, which can be understood as an increase in personal resources within the framework of the JD-R model. Psychological resilience reduces burnout by enabling individuals to manage stress more effectively. Thus, transformational leadership enhances psychological resilience and contributes to lowering burnout levels. Indeed, [18] ([18]) found that transformational leadership significantly reduces burnout, with employee resilience serving as a mediating factor. Similarly, [70] ([70]) demonstrated that transformational leadership and psychological empowerment negatively predict teacher burnout and that psychological empowerment mediates the relationship between transformational leadership and burnout. Based on this, we propose the following hypothesis:

**H10:** *Psychological resilience mediates the relationship between transformational leadership and burnout*.

## 4. Materials and Methods

Drawing upon the theoretical framework and empirical findings reviewed above, the following section outlines the research design and methodology employed to investigate the proposed hypotheses within the context of the teaching profession. The research model is given in Figure 1. The model illustrates the effects of three different leadership styles—servant, authentic, and transformational—on burnout through the mediating role of psychological resilience. It examines how each leadership style contributes to enhancing employees’ psychological resilience and, in turn, how this resilience influences burnout levels.

### 4.1. Procedure

The population of this study consists of teachers affiliated with the union. To administer the survey to the sample group, the necessary ethical approval was obtained from Mersin University, Turkiye (Ethics No: 385). Subsequently, a meeting was requested with the union president, during which information about the research was provided, and permission was requested to collect data. After receiving approval, a meeting was held with the personnel manager. On 11 January 2025, the survey link was sent to the union members. Once the participants completed the survey, these data were directly sent to the researchers’ accounts. Participants were informed on the first page of the online survey. The information provided emphasized that participation was voluntary, these data would not be shared with third parties, and anonymity was ensured. The data collection process was concluded after two weeks, as it was determined that the data flow had ceased. During the meeting with the union official, it was noted that the survey was distributed to 1211 participants.

The 425 survey responses obtained were examined by the researchers, and 38 responses were excluded from this study because of being incomplete or incorrectly filled. As a result, this study was based on 387 valid responses (response rate: 31.96%).

### 4.2. Measures

Well-structured scales that have been frequently used in the past years and whose reliability and validity have been tested in Turkish were used as data collection tools. All scales are rated from 1 to 5.

The 7-item “Servant Leadership Scale” developed by [48] ([48]) and adapted to Turkish by [43] ([43]) was used. Sample items on the scale are as follows: “I would seek help from my manager if I had a personal problem”, and “My manager can tell if something is going wrong”.

The 16-item “Authentic Leadership Scale” developed by [74] ([74]) and adapted to Turkish by [67] ([67]) was used. Sample items on the scale are as follows: “Seeks feedback to improve interactions with others”, and “Says exactly what he or she means”.

The 8-item “Authentic Leadership Scale” developed by [14] ([14]) and adapted to Turkish by [60] ([60]) was used. Sample items on the scale are as follows: “(He/she develops ways of motivating us”, and “(He/she gets us to rely on reasoning and evidence to solve problems”.

The 10-item “Brief Resilience Scale” developed by [64] ([64]) and adapted to Turkish by [24] ([24]) was used. Sample items on the scale are as follows: “I tend to bounce back quickly after hard times”, and “3. It does not take me long to recover from a stressful event”.

The 10-item “Burnout Measure Short Version” developed by [54] ([54]) and adapted to Turkish by [20] ([20]) was used. Sample items on the scale are as follows: “I feel worthless and like a failure”, and “I feel let down by people”.

## 5. Statistical Analysis

In this study, analyses were first conducted to assess common method bias (CMB), followed by tests for the reliability and validity of the constructs. Subsequently, hypothesis testing was carried out using Process Macro (Model 4). The analyses were performed using AMOS 26.0 and SPSS 27.0 software. Based on data collected and analyzed through the procedures outlined above, the results of the statistical analyses are presented below to evaluate the validity of the proposed research model and hypotheses.

## 6. Results

### 6.1. Samples

Descriptive statistics regarding demographic data for the sample group included in this study are given in Table 1.

An analysis of the participants’ demographic information reveals that 248 (64.08%) are female, while 139 (35.92%) are male. The majority of the participants (39.02%) are aged between 35 and 44. Additionally, 80.62% of the participants hold a bachelor’s degree, and 37.73% have more than 20 years of work experience.

### 6.2. Common Method Bias

A variety of procedural methods were employed to minimize the issue of common method bias. First, previously developed and well-established scales were utilized to avoid ambiguity in the statements. The clarity of the items in the scales was confirmed through a pilot survey prior to administering the questionnaire to the sample population. To reduce cognitive fatigue among participants, the questionnaire was kept concise, and a briefing was provided on the first page of the survey. The information included assurances regarding data confidentiality, the voluntary nature of participation, the absence of right or wrong answers, and the importance of participants’ opinions, with a commitment that these data would not be shared with third parties.

In addition to the procedural methods, common method bias (CMB) was assessed using statistical techniques commonly employed in recent studies. First, Harman’s single-factor test was conducted. The analysis revealed that no single factor was dominant, and the highest explained variance was 31.11%. This value is below the lower threshold recommended in the literature ([28]). Second, a correlation analysis was performed, and the correlation coefficients (r) between the variables were found to be below 0.90 ([12]). Finally, variance inflation factor (VIF) values were examined, and it was determined that all VIF values were below 3.3. Based on these results and the procedures applied previously, it was concluded that CMB does not pose a significant issue in this study ([66]; [45]).

### 6.3. Preliminary Checks

Before testing the hypotheses, the reliability and validity of the constructs in the research were assessed. For reliability evaluation, internal consistency (Cronbach’s Alpha), composite reliability (CR), and McDonald’s Omega reliability coefficients were considered. In the validity tests, both convergent and discriminant validity were examined.

In order to assess the reliability of the constructs, the indicator values (Cronbach’s alpha, CR, and Omega coefficients) should be 0.7 or above ([32]; [66]). The analysis results show that Cronbach’s alpha values range from 0.897 to 0.971, CR values range from 0.899 to 0.971, and Omega coefficients range from 0.898 to 0.971. The fact that these values are 0.7 or above confirms that the related constructs are internally consistent and reliable.

AVE and CR values were checked for convergent validity. According to the literature, for convergent validity, the average variance extracted (AVE) value should be greater than 0.5 and should be lower than the composite reliability (CR) value ([44]). The findings presented in Table 2 show that all these criteria are met, confirming that the constructs have convergent validity.

For discriminant validity, the Heterotrait–Monotrait Ratio (HTMT) criteria proposed by [34] ([34]) and the criteria proposed by [27] ([27]) were used. The HTMT test, as proposed by [34] ([34]), is calculated based on the geometric mean of the average correlation between constructs and the average correlation of items within the same construct. According to the researchers, heterotrait correlations should be smaller than monotrait correlations, meaning the HTMT ratio should be below 1.0. [34] ([34]) also suggest that discriminant validity can be considered when the HTMT value is below 0.85 for concepts that are closely related and below 0.90 for concepts that are not closely related. The results of the analysis are presented in Table 3.

The results in Table 3 show that the HTMT values are within the threshold limits recommended by [34] ([34]).

Another method frequently used in the literature to assess discriminant validity is the criteria proposed by [27] ([27]). According to this method, the square root of the AVE (√AVE) value should be higher than the correlation values with other variables. The results of the analyses conducted according to the [27] ([27]) criteria are presented in Table 4.

As seen in Table 4, the √AVE value is higher than the correlation values with other variables. Thus, both the criteria of [34] ([34]) and the criteria proposed by [27] ([27]) are met. All these findings confirm that the constructs have both convergent and discriminant validity.

## 7. Hypothesis Tests

Process Macro v4.2 (Model 4) was used to test the research hypotheses. The results of the analyses, conducted at a 95% confidence interval with 5000 resamples, are presented in Table 5.

The results in Table 5 show that servant leadership (β = 0.284, *p* < 0.05, 95% CI [0.171, 0.397]), transformational leadership (β = 0.256, *p* < 0.05, 95% CI [0.156, 0.346]), and authentic leadership (β = 0.214, *p* < 0.05, 95% CI [0.104, 0.325]) have a positive effect on psychological resilience. Thus, Hypotheses 1, 2, and 3 are supported.

The results also show that servant leadership (β = −0.278, *p* < 0.05, 95% CI [−0.405, −0.168]), transformational leadership (β = −0.287, *p* < 0.05, 95% CI [−0.414, −0.142]), and authentic leadership (β = −0.302, *p* < 0.05, 95% CI [−0.428, −0.175]) have a negative effect on burnout. Additionally, the effect of psychological resilience on burnout is also negative (β = −0.736, *p* < 0.05, 95% CI [−0.882, −0.590]). Thus, Hypotheses 4, 5, 6, and 7 are supported.

The final hypothesis of this study aims to determine the mediating role of psychological resilience in the effect of servant leadership, transformational leadership, and authentic leadership on burnout. To test this, the indirect effect was examined. The results of the analysis confirm that psychological resilience mediates the effect of servant leadership (β = −0.209, *p* < 0.05, 95% BCA CI [−0.316, −0.112]), transformational leadership (β = −0.183, *p* < 0.05, 95% BCA CI [−0.287, −0.094]), and authentic leadership (β = −0.160, *p* < 0.05, 95% BCA CI [−0.429, −0.175]) on burnout. Therefore, Hypotheses 8, 9, and 10 are supported.

## 8. Discussion and Conclusions

The present study aimed to examine the effects of positive leadership styles—namely servant, authentic, and transformational leadership—on employees’ psychological resilience and burnout levels. In addition, this study sought to explore the mediating role of psychological resilience in the relationship between positive leadership styles and burnout.

### 8.1. Discussion

This study aimed to examine the relationships between positive leadership styles—specifically servant, authentic, and transformational leadership—and psychological resilience among teachers, as well as the mediating role of psychological resilience in reducing burnout. The findings offer valuable insights into how leadership practices can strengthen teachers’ psychological resources, particularly within the Turkish cultural and educational context.

The results revealed that transformational and servant leadership were significantly associated with higher levels of psychological resilience. This finding aligns with previous research suggesting that these leadership styles enhance emotional strength and coping mechanisms in high-stress professions such as teaching ([26]; [1]). Within the Turkish context, where hierarchical structures remain prevalent in many school environments, the presence of servant or transformational leaders may serve as a critical buffer against burnout by fostering a sense of empowerment and support among teachers.

Interestingly, although authentic leadership also demonstrated a positive relationship with psychological resilience, its impact appeared to be somewhat weaker. This may be attributed to the contextual sensitivity of authenticity as a leadership construct, which may operate differently in collectivist cultures such as Turkiye, where relational harmony and indirect communication are highly valued ([11]).

The mediating role of psychological resilience between leadership and burnout further supports the assumptions of the Job Demands–Resources (JD-R) model, which underscores the protective function of both personal and job-related resources. Teachers who feel supported by their leaders are more likely to develop the internal resilience needed to withstand chronic stressors such as emotional labor, role ambiguity, and institutional pressures. This finding underscores the importance of integrating psychological resource development into teacher training and school leadership programs in Turkiye.

From a cultural perspective, the emphasis on leader—employee relationships—particularly those marked by empathy, trust, and individualized support—aligns closely with the collectivist orientation of Turkish society. In such contexts, relational leadership styles may exert a stronger influence on psychological outcomes than task-oriented or transactional approaches.

### 8.2. Theoretical Contribution

The results of the analysis revealed a significant relationship between servant leadership and psychological resilience. Previous studies on this subject have consistently found positive associations between servant leadership and resilience ([58]). Similarly, the analysis also identified a significant relationship between authentic leadership and psychological resilience. This finding aligns with the existing literature, which highlights the positive impact of authentic leadership on enhancing resilience in educational contexts ([29]; [55]).

The analysis results revealed a positive and significant relationship between transformational leadership and psychological resilience. This finding is consistent with general trends observed in the literature. Previous studies have highlighted a positive relationship between transformational leadership and psychological empowerment. [1] ([1]) and [40] ([40]) found that transformational leadership enhances employees’ overall resilience levels. It can be inferred that transformational leaders’ inspirational, motivational, and supportive approaches strengthen employees’ self-efficacy, improve their ability to cope with challenges, and generally increase their psychological resilience. Furthermore, existing research indicates that individuals with high psychological resilience tend to experience lower levels of burnout ([4]). The findings of this study support this theoretical framework. Specifically, the results confirm that teachers with higher psychological resilience exhibit fewer burnout symptoms and are better equipped to manage work-related stress.

Studies have shown that servant leadership reduces burnout ([53]; [3]; [5]). The findings from the current study align with the literature. Specifically, a statistically significant relationship was found between authentic leadership and burnout. This result is consistent with other studies ([46]; [57]), which suggest that authentic leadership plays a role in reducing teachers’ burnout levels.

Studies show that there is a negative relationship between transformational leadership perception and burnout levels ([35]; [70]; [69]). In this study, an inverse and significant relationship was found between transformational leadership and burnout. The effect of transformational leadership may vary depending on factors such as organizational culture, personality traits of employees, and the stress level of the work environment. For example, [35] ([35]) reported that transformational leadership reduces employees’ emotional exhaustion levels, but this effect is more pronounced in high-stress work environments. Finally, this study examined the impact of servant, authentic, and transformational leadership styles on teacher burnout and the mediating role of psychological resilience in this relationship. Based on the Job Demands–Resources (JD-R) model, it is assumed that leadership styles can be considered important resources in the work environment and that these resources can reduce burnout by increasing the psychological resilience of teachers. The findings of this study revealed that psychological resilience has a partial mediating effect on the relationship between servant, authentic, and transformational leadership styles and burnout. This result suggests that teachers with high psychological resilience can reduce the risk of burnout by using the support they receive from their leaders more effectively. This finding is also supported by other studies in the literature. For example, [76] ([76]) found that employee resilience played a mediating role in the positive effect of servant leadership on work engagement. Similarly, [16] ([16]) revealed the mediating effect of employee resilience in increasing employee work engagement through servant leadership. Additionally, [18] ([18]) found that transformational leadership significantly reduces burnout, and employee resilience plays a mediating role in this relationship. Regarding authentic leadership, [2] ([2]) found that employees’ psychological capital (psychological resilience is a part of this capital) mediates the relationship between authentic leadership and burnout. [42]’s ([42]) research also showed that authentic leadership has positive effects on employees’ positive psychological capital, which in turn reduces job burnout.

This study revealed that servant, authentic, and transformational leadership styles have a direct, statistically significant relationship with burnout and psychological resilience. Moreover, mediation analyses indicated that these leadership styles may indirectly reduce burnout levels by enhancing teachers’ psychological resilience. This finding suggests that the impact of leadership styles is not simple or direct but rather emerges through a more complex process. Additionally, previous studies have generally addressed the relationships between leadership styles, psychological resilience, and burnout separately. This study fills the gap in the literature by examining the mediating role of psychological resilience in the effect of leadership styles (servant, authentic, and transformational) on burnout, considering all three variables together. This holistic approach reveals that leadership styles may have an indirect effect on burnout by increasing teachers’ psychological resilience rather than a direct effect on burnout. It makes a unique contribution to the literature by demonstrating that these relationships are more complex and may interact with other factors. Therefore, this study provides an important foundation for future research by exploring the interactions between leadership, psychological resilience, and burnout from a more comprehensive perspective.

### 8.3. Practical Implications

The findings of this study have significant practical implications for efforts to cope with burnout and enhance resilience in the workplace. First, the fact that leadership styles alone do not have a direct effect on burnout suggests that this issue is multidimensional and complex. Leadership styles can indirectly affect burnout by increasing teachers’ psychological resilience. This highlights the need for leadership training programs that educate leaders on how to support teachers’ psychological resilience rather than focusing solely on specific leadership styles. In particular, it is crucial to encourage leadership behaviors that help teachers feel valued, build confidence, develop skills to cope with challenges and find meaningful purpose. Additionally, providing training and psychological counseling services for teachers to enhance their stress management skills can be an effective method for addressing burnout. Managers must be aware of such support mechanisms and direct their subordinates to these resources. Finally, it is important to recognize that organizational culture also plays a critical role. Creating a culture that promotes psychological safety at work, encourages open communication, and values employee feedback is essential for preventing burnout.

The managerial implications of this research call for organizations to re-evaluate their leadership development programs and employee support strategies. Leadership training should focus on strategies to enhance employees’ psychological resilience. Managers should aim to help employees strengthen their self-efficacy, increase their emotional resilience, foster social support, and develop stress-coping skills. Moreover, managers should be mindful of how their leadership style impacts employees’ resilience and burnout levels. Organizations should not only evaluate leaders based on performance outcomes but also consider their impact on employees’ psychological well-being within leadership evaluation systems. Furthermore, identifying employees exhibiting burnout symptoms and developing support programs for them can help organizations protect employee health and improve productivity. In light of these findings, managers should adopt a more holistic approach to leadership practices and implement strategies that support employee well-being.

### 8.4. Limitations and Future Direction of This Study

This study has several limitations. First, the cross-sectional nature of these data makes it challenging to draw definitive causal conclusions about the relationships between leadership styles, psychological resilience, and burnout. Future research should employ longitudinal studies to better understand these relationships and explore how they evolve over time, as well as investigate potential causal effects in greater depth. Second, the data collection method relied on self-reporting, which may introduce biases such as social desirability and memory errors, potentially influencing participants’ responses. Third, this study was conducted on a specific sample, which limits the generalizability of the results to other sectors or cultural contexts. Future studies should seek to expand the generalizability of their findings by examining the relationships between these variables across different sectors and cultural contexts. Additionally, other factors (e.g., organizational climate, job stress levels, personal characteristics, etc.) that may mediate or moderate the relationships between the variables studied were not considered. Future research could benefit from exploring these factors to gain a more comprehensive understanding of the complex dynamics at play between leadership, psychological resilience, and burnout. Considering these limitations, more extensive and multifaceted studies will contribute to a deeper understanding of the relationships among leadership, psychological resilience, and burnout and help to develop more effective practices in this area.

## Figures and Tables

**Figure 1 behavsci-15-00713-f001:**
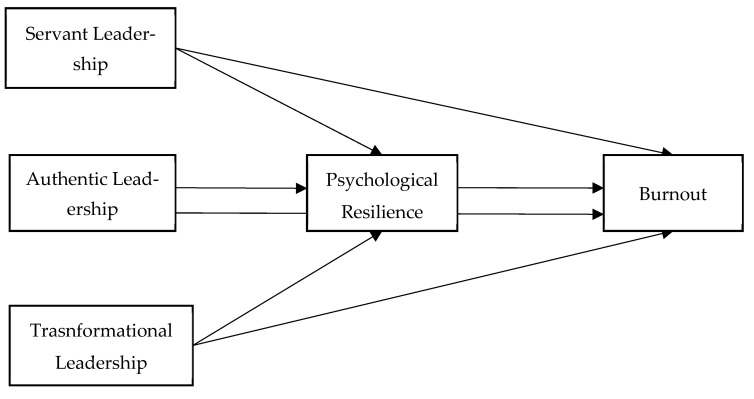
Research Model.

**Table 1 behavsci-15-00713-t001:** Demographic information of participants.

Variable	Category	N	Percentage (%)
Gender	Man	139	35.92
	Woman	248	64.08
Age	18–24	16	4.13
	25–34	61	15.76
	35–44	151	39.02
	45–54	94	24.29
	55 and over	65	16.80
Education	Bachelor’s Degree	312	80.62
	Master’s Degree	70	18.09
	Doctorate	5	1.29
Length of service in the institution	Less than 5 years	183	47.29
	5–9	85	21.96
	10–14	56	14.47
	15–19	22	5.68
	20 and over	41	10.59
Experience (Years)	Less than 5 years	54	13.95
	5–9	41	10.59
	10–14	78	20.16
	15–19	68	17.57
	20 and over	146	37.73

**Table 2 behavsci-15-00713-t002:** Measurement Model.

Variables	Statements	Factor Loadings	Cronbach’s Alfa	CR	McDonald’s Omega	AVE
Servant Leadership	SL1	0.741	0.911	0.914	0.915	0.609
	SL2	0.862				
	SL3	0.710				
	SL4	0.902				
	SL5	0.888				
	SL6	0.708				
	SL7	0.603				
Transformational Leader	TL1	0.837	0.971	0.971	0.971	0.805
	TL2	0.895				
	TL3	0.895				
	TL4	0.834				
	TL5	0.859				
	TL6	0.944				
	TL7	0.963				
	TL8	0.945				
Authentic Leadership	A1	0.773	0.966	0.967	0.967	0.652
	A2	0.832				
	A3	0.791				
	A4	0.524				
	A5	0.833				
	A6	0.923				
	A7	0.718				
	A8	0.799				
	A9	0.571				
	A10	0.635				
	A11	0.865				
	A12	0.897				
	A13	0.934				
	A14	0.929				
	A15	0.918				
	A16	0.824				
Psychological Resilience	PR1	0.764	0.897	0.899	0.898	0.602
	PR1	0.815				
	PR1	0.634				
	PR1	0.815				
	PR1	0.699				
	PR1	0.899				
Burnout	B1	0.672	0.935	0.937	0.939	0.603
	B2	0.729				
	B3	0.841				
	B4	0.851				
	B5	0.845				
	B6	0.912				
	B7	0.754				
	B8	0.703				
	B9	0.501				
	B10	0.870				

**Table 3 behavsci-15-00713-t003:** Discriminant Validity (HTMT Criterion).

	Servant Leadership	Transformational Leader	Authentic Leadership	Psychological Resilience
Transformational Leader	0.830			
Authentic Leadership	0.765	0.827		
Psychological Resilience	0.339	0.333	0.256	
Burnout	0.409	0.431	0.388	0.652

**Table 4 behavsci-15-00713-t004:** Discriminant Validity (Fornell and Larcker Criterion).

	1	2	3	4	5
Servant Leadership	0.780 ^a^				
Transformational Leader	0.803 **	0.897 ^a^			
Authentic Leadership	0.753 **	0.801 **	0.808 ^a^		
Psychological Resilience	0.346 **	0.305 **	0.231 **	0.776 ^a^	
Burnout	−0.386 **	−0.404 **	−0.378 **	−0.592 **	0.777 ^a^

** *p* < 0.05, ^a^: √AVE.

**Table 5 behavsci-15-00713-t005:** Direct and indirect effects of each model path.

Estimate	Paths	β	SE	95%CI	
				LLCI	ULCI
Direct effect	SL→ PR	0.284	0.057	0.171	0.397
	TL→ PR	0.256	0.053	0.156	0.346
	AL→ PR	0.214	0.056	0.104	0.325
	SL→ B	−0.278	0.069	−0.405	−0.168
	TL→ B	−0.287	0.061	−0.414	−0.142
	AL→ B	−0.302	0.064	−0.428	−0.175
	PR→ B	−0.736	0.074	−0.882	−0.590
Indirect effect	SR→PR→B	−0.209	0.051	−0.316	−0.112
	TL→PR→B	−0.183	0.048	−0.287	−0.094
	AL→PR→B	−0.160	0.052	−0.429	−0.175

SL: Servant Leadership, TL: Transformational Leadership, AL: Authentic Leadership, PR: Psychological Resilience, B: Burnout.

## Data Availability

The data presented in this study are available on request from the corresponding author.

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
