# Peer review of "Relationships Between Positive Leadership Styles, Psychological Resilience, and Burnout: An Empirical Study Among Turkish Teachers"

_behavsci, 2025, doi:10.3390/bs15060713_

Round 1
Reviewer 1 Report
Comments and Suggestions for Authors
Your paper tackles a timely and underexplored topic, and you've laid down a solid empirical foundation—it’s clear you’ve put serious thought and effort into the research design and data presentation. That said, there are some areas that could use tightening. The theoretical framing feels a bit light; pulling in more recent or foundational scholarship would help anchor your work in the larger conversation and show why it matters right now. Structurally, some of the transitions—especially between the lit review, methods, and findings—are a little choppy. Guiding the reader more smoothly would really elevate the flow. The writing itself is generally clear, but a language polish would go a long way—some phrasing feels clunky or imprecise, and that can distract from the strong ideas you’re trying to communicate. Lastly, the discussion section has room to stretch. With such culturally specific insights, you’ve got a great opportunity to dig deeper into the implications. If you make these tweaks, the paper could land with more impact and be a real contribution to the field.
Comments on the Quality of English LanguageThe English is generally clear, and the core ideas come through—but there are a few rough edges. Some sentences feel awkward or overly complicated, and a few grammar glitches pop up here and there. It’s nothing major, but a careful line edit would help smooth things out and let your ideas shine without distraction. A bit of polish could really sharpen the delivery.
Author Response
RESPONSE TO REVIEWER 1
We would like to extend our sincere appreciation to you for your comprehensive evaluations and valuable feedback. Your insightful comments and constructive suggestions have been instrumental in enhancing the conceptual rigor, contextual depth, and overall clarity of our manuscript. We have carefully addressed each of the points raised, implementing substantial revisions to strengthen the theoretical framework, refine the interpretation of findings, and improve the presentation throughout. We are grateful for the time and expertise you dedicated to reviewing our work and for your contributions to improving its scholarly quality.
Comment 1:
“Your paper tackles a timely and underexplored topic, and you’ve laid down a solid empirical foundation—it’s clear you’ve put serious thought and effort into the research design and data presentation. That said, there are some areas that could use tightening. The theoretical framing feels a bit light; pulling in more recent or foundational scholarship would help anchor your work in the larger conversation and show why it matters right now.”
Response:
Thank you for your constructive feedback. In line with your suggestion, we have revised the theoretical framework to include both recent and foundational studies, particularly focusing on the JD-R and COR models in educational contexts. These updates aim to place our study more firmly within the existing literature and highlight its current relevance. The revisions can be found in Section 2 and in the final paragraph of the Introduction.
Comment 2:
“Structurally, some of the transitions—especially between the lit review, methods, and findings—are a little choppy. Guiding the reader more smoothly would really elevate the flow.”
Response:
Thank you for this helpful observation. To improve the overall flow of the manuscript, we have added transitional sentences at the end of the literature review and at the beginning of the methods section. These changes aim to provide a smoother and more logical progression between sections. The revisions can be found at the end of Section 3 and the start of Section 5.
Comment 3:
“The writing itself is generally clear, but a language polish would go a long way—some phrasing feels clunky or imprecise, and that can distract from the strong ideas you’re trying to communicate.”
Response:
Thank you for pointing this out. We have reviewed the manuscript to improve clarity and ensure a more consistent academic tone. Several sentences have been rephrased to enhance readability and precision, especially in the Introduction, Conceptual Framework, and Discussion sections.
Comment 4:
“Lastly, the discussion section has room to stretch. With such culturally specific insights, you’ve got a great opportunity to dig deeper into the implications.”
Response:
Thank you for this valuable suggestion. We have expanded the Discussion to engage more deeply with the cultural dimensions and to elaborate on the broader implications of the findings. A new and more comprehensive Discussion section has been added as Section 8.
Comment 5
The English is generally clear, and the core ideas come through, but there are a few rough edges. Some sentences feel awkward or overly complicated, and a few grammar glitches pop up here and there. It’s nothing major, but a careful line edit would help smooth things out and let your ideas shine without distraction. A bit of polish could really sharpen the delivery.
Response:
In response to your comments, we have carefully reviewed and revised the manuscript to improve the clarity and fluency of the English. We have addressed awkward or overly complex sentences and corrected minor grammatical issues throughout the text.
Reviewer 2 Report
Comments and Suggestions for Authors
The paper addresses an intriguing and original topic, but certain improvements are necessary:
- In Section 1, Introduction, the research gap is not identified.
- In Section 2, Conceptual Framework, the authors fail to contextualize their research.
- The basis for the research hypotheses is weak. It is advisable to strengthen them, particularly H1-H3.
- There is no discussion section.
Additional comments:
The paper aims to examine the effects of positive leadership approaches on the psychological resilience and burnout levels of employees;
- The topic is original and relevant; however, the authors do not identify the research gap.
- The authors' contributions compared to other published materials are not very clearly highlighted since the paper does not have a discussion section. Additionally, subsection 8.1 of the Conclusions does not present the theoretical contributions, but the authors relate their results to those of other materials. These comparisons do not reveal the elements added by the authors to the subject.
- The methodology requires more rigorous substantiation for hypotheses H1 and H2.
- The first part of the Conclusions section should be moved to a discussion section.
- The references are appropriate.
Author Response
RESPONSE TO REVIEWER 2
We would like to extend our sincere appreciation to you for your comprehensive evaluations and valuable feedback. Your insightful comments and constructive suggestions have been instrumental in enhancing the conceptual rigor, contextual depth, and overall clarity of our manuscript. We have carefully addressed each of the points raised, implementing substantial revisions to strengthen the theoretical framework, refine the interpretation of findings, and improve the presentation throughout. We are grateful for the time and expertise you dedicated to reviewing our work and for your contributions to improving its scholarly quality.
Comment 1:
“In Section 1, Introduction, the research gap is not identified.”
Response:
Thank you for this important observation. We have revised the Introduction to more clearly define the research gap, with a particular focus on the limited number of integrative studies examining the mediating role of psychological resilience in the relationship between leadership and burnout within the teaching profession.
Comment 2:
“In Section 2, Conceptual Framework, the authors fail to contextualize their research.”
Response:
Thank you for this helpful comment. In response, we have revised the Conceptual Framework to more clearly situate each key construct—leadership theories, psychological resilience, and burnout—within the context of the teaching profession. These changes aim to strengthen the contextual relevance of the study. Please refer to Sections 2.1 to 2.5 (pages 3–6) for the updated content.
Comment 3:
“The basis for the research hypotheses is weak. It is advisable to strengthen them, particularly H1-H3.”
Response:
Thank you for this valuable suggestion. We have strengthened the theoretical foundations of Hypotheses 1 through 3 by incorporating more robust literature support and clearer theoretical reasoning. In particular, we have elaborated on the links between servant, authentic, and transformational leadership and teacher resilience. Please see Sections 3.1 to 3.3 for these revisions.
Comment 4:
“There is no discussion section.”
Response:
Thank you for your observation. As noted in our earlier response to Reviewer 1, we have added a new Discussion section (Section 8). This section provides a detailed interpretation of the findings, compares them with existing literature, and discusses their implications for educational practice.
Reviewer 3 Report
Comments and Suggestions for Authors
Although the research issue is not original and recent the current work respects writing standards and academic structure. It is clear and coherent/consistent. The arguments are solidly supported and they reinforcing the conclusion. The literature and methodological procedures are also well described.
It is only suggested that the title should include the empirical field (country and professional group surveyed).
Author Response
RESPONSE TO REVIEWER 3
We sincerely thank the reviewer for the insightful comment regarding the title. The suggestion to include the empirical context—specifically the country and professional group surveyed—was very helpful in clarifying the scope of our study. We appreciate your thoughtful input, which has contributed to strengthening the clarity and relevance of our manuscript.
Comment 1:
“It is only suggested that the title should include the empirical field (country and professional group surveyed).”
Response:
Thank you for the suggestion. In response, we have revised the title to more clearly indicate the empirical scope of the study, including the country and professional group surveyed.
Reviewer 4 Report
Comments and Suggestions for Authors
Brief summary:
The article analyzes the mediating effect of the psychological resilience of leadership styles on teachers' burnout. The study addresses an important and actual subject regarding school actors, filling a research gap in the literature. A coherent theoretical framework supports the article, the methodological design is adequate, and the hypothesis testing was conducted professionally. After a few upgrades and improvements, the manuscript can provide relevant knowledge on educational leadership and teacher burnout.
Writing:
The message is clearly understood, but the English can be slightly refined for clarity and flow.
General concept comments:
The article asserts a relevant research topic in educational sciences, providing impactful information for policymakers, scholars, school principals, and teachers.
The abstract clarifies the study’s purpose statement, the sample/participants, the study variables, the data-gathering instruments, and the main findings. However, it lacks highlights on the guiding theory, the research gap (too vague), and aspects of methods, such as data analysis strategy. The study regards educational contexts, but it repeatedly refers to ‘employees,’ and the theoretical approach presented in the first seven lines (16 to 21) regards business contexts, leading to some confusion. Considerer substituting the word “employee” and reducing the enterprise context. The referred text (lines 16 to 21) is a repetition of the first paragraph of the introduction. You can mobilize readers more if you focus on the educational context. Consider also being more concise regarding so. Most of the theories/references you call regarding the conceptual framework belong to non-educational fields of knowledge, suggesting a gap concerning the theme from your research within the literature. Consider valuing your approach with this idea (not mandatory). I believe that is why your manuscript is so interesting.
Consider, when possible, substituting the word employees for teachers all over the manuscript (primarily when referring to the study design, data gathering, results, findings,…).
Conceptual framework: consider organizing the text in structured and well-defined subsections concerning the three leadership styles. It would help clarify the flow of ideas and make the content easier for readers.
From my point of view, “3. Interconceptual Relations and Hypotheses” (line 196) is a subsection of the framework.
The hypotheses are well-defined, relevant, and clearly supported by the theoretical evidence presented. However, the research model in Figure 1 (page 9) does not consider hypothesis 7. Please readjust the model. On page 9, line 398, explain the model briefly (what it represents, or summarize it).
The statistical analysis design is not described enough. Explain how reliability and validity constructs were tested – which measures were used, such as Cronbach’s alpha, or factor analysis… Since the survey might be applied to a different population group, the authors should explain why a pilot study on validation of the survey can be neglected in the present study.
Process Macro Model is an adequate statistical method for your study. However, you should provide information/justify it based on what you want to test – the mediating effect of psychological resilience and that it allows you to estimate both direct and indirect effects concerning your variables. Please explain your methodological option. Provide some information regarding the statistics you choose.
Regarding the methodology, inform that the questionnaire included some questions to clarify some participants' demographic features. The methodology also does not mention the pilot survey, which is essential regarding the study's validity. Please complete the method with the referred elements. You refer to the pilot survey in the results section, but nothing is clarified in the methods section.
The results are presented with clairvoyance, and the discussion suits the study's aim.
Specific comments:
On page 2, lines 61 and 62 - “In the literature, it is emphasized that leaders play a critical role in 61 reducing employees’ burnout levels” – provide references.
On page 2, in line 67, consider substituting the word ‘effects’ because you are studying an association between variables.
On page 2, line 68, servant, transformational, and authentic leadership are styles instead of approaches, even though they may represent a positive leadership approach. You should refer to ‘approach’ if the focus is the philosophy or framework behind the behavior and to ‘style’ if you emphasize the observable behaviors or traits the leader exhibits. Consider uniformizing the writing all over the text, applying this criterion.
On page 3, lines 93 and 94, - “While improvement represents an approach that encourages continuous improvement…” – is unclear. The sentence evidence circularity and lack of specificity because it uses “improvement” to define “improvement” and does not clarify the concept for the reader. What kind of improvement is it referring to?
On page 3, lines 115 to 119 – The sentence is too long and hard to understand. What “they” meant/refers to is not comprehensible.
On page 4, lines 157 to 160 – The sentence is long; breaking or restructuring it will improve readability.
On page 4, lines 167 to 170 - The repetition of "them" and "their" in the sentence (especially so close together) can introduce some mild ambiguity or at least slow down the reader as they mentally track who "them" and "their" refer to.
On page 7, line 334, when you wrote, “In this respect, leadership styles can be considered as important job resources in the business environment,” consider expanding the idea to the school organization.
On page 14, line 540, the section 8 uses capital letters. Harmonize according to the other sections. Section 8 is more than just conclusions; consider redefining and adopting ‘discussion and conclusions’.
On Page 15, line 564, you refer to ‘individuals’. Your study was done with teachers, so I think you should emphasize that. All your findings concern the teaching profession and educational leadership. Apply this suggestion throughout the discussion of results.
On page 15, line 579, a ‘period’ is missing in the sentence. Consider starting a new paragraph.
Comments on the Quality of English LanguageThe message is clearly understood, but the English can be slightly refined for clarity and flow. A more concise and direct approach will fill that purpose.
Author Response
RESPONSE TO REVIEWER 4
We would like to extend our sincere appreciation to you for your comprehensive evaluations and valuable feedback. Your insightful comments and constructive suggestions have been instrumental in enhancing the conceptual rigor, contextual depth, and overall clarity of our manuscript. We have carefully addressed each of the points raised, implementing substantial revisions to strengthen the theoretical framework, refine the interpretation of findings, and improve the presentation throughout. We are grateful for the time and expertise you dedicated to reviewing our work and for your contributions to improving its scholarly quality.
Comment 1
“The abstract clarifies the study’s purpose statement, the sample/participants, the study variables, the data-gathering instruments, and the main findings. However, it lacks highlights on the guiding theory, the research gap (too vague), and aspects of methods, such as data analysis strategy. The study regards educational contexts, but it repeatedly refers to ‘employees,’ and the theoretical approach presented in the first seven lines (16 to 21) regards business contexts, leading to some confusion. Considerer substituting the word “employee” and reducing the enterprise context. The referred text (lines 16 to 21) is a repetition of the first paragraph of the introduction. You can mobilize readers more if you focus on the educational context. Consider also being more concise regarding so. Most of the theories/references you call regarding the conceptual framework belong to non-educational fields of knowledge, suggesting a gap concerning the theme from your research within the literature. Consider valuing your approach with this idea (not mandatory). I believe that is why your manuscript is so interesting.”
Response:
Thank you for your thoughtful and constructive feedback. Regarding the terminology, we have retained the use of “employees” in the main text as per your suggestion, while ensuring the abstract explicitly clarifies that the study specifically focuses on private school teachers in Türkiye. This maintains consistency with your recommendation while aligning the abstract with the study’s context.
We have also strengthened the articulation of the research gap by more clearly highlighting the lack of integrated studies on positive leadership styles, resilience, and burnout in educational settings, especially within non-Western contexts. Furthermore, we have added methodological details, including the cross-sectional design and the use of PROCESS Macro (Model 4) for mediation analysis, to the abstract to improve clarity and transparency.
Although our theoretical framework incorporates organizational theories (LMX, COR, and JD-R), we have emphasized their relevance to educational contexts by citing relevant adaptations in the teacher well-being literature. These revisions aim to clarify the educational focus of the study and address the concerns you raised regarding the application of business-oriented theories.
We believe these improvements have significantly enhanced the clarity, theoretical grounding, and contribution of the manuscript, particularly in relation to understanding leadership’s impact on educator well-being.
Comment 2
“Conceptual framework: consider organizing the text in structured and well-defined subsections concerning the three leadership styles. It would help clarify the flow of ideas and make the content easier for readers.”
Response:
Thank you for this helpful suggestion. In response, we have reorganized the section on the conceptual framework, introducing clearly structured subsections for each of the three leadership styles. This revision aims to improve the clarity and flow of ideas, making the content more accessible for readers.
Comment 3
“The hypotheses are well-defined, relevant, and clearly supported by the theoretical evidence presented. However, the research model in Figure 1 (page 9) does not consider hypothesis 7. Please readjust the model. On page 9, line 398, explain the model briefly (what it represents, or summarize it).”
Response:
Thank you for your valuable feedback. In response, we have revised Figure 1 (page 9) to include Hypothesis 7 in the research model. Additionally, we have added a brief explanation of the model on line 415 to clarify its representation and summarize its key components.
Comment 4
“On page 2, lines 61 and 62 - “In the literature, it is emphasized that leaders play a critical role in 61 reducing employees’ burnout levels” – provide references.”
Response:
Thank you for your observation. A relevant reference has been added to support this statement on page 2, lines 61 and 62.
Comment 5
“On page 2, in line 67, consider substituting the word ‘effects’ because you are studying an association between variables.”
Response:
Thank you for your insightful suggestion. As recommended, we have replaced the word “effects” with “relationships” to more accurately reflect the associative nature of the study, avoiding any implication of causality. The revised sentence can now be found on page 2, lines 70–74.
Comment 6
“On page 2, line 68, servant, transformational, and authentic leadership are styles instead of approaches, even though they may represent a positive leadership approach. You should refer to ‘approach’ if the focus is the philosophy or framework behind the behavior and to ‘style’ if you emphasize the observable behaviors or traits the leader exhibits. Consider uniformizing the writing all over the text, applying this criterion.”
Response:
Thank you for this insightful comment. We agree with your distinction between ‘style’ and ‘approach.’ Since our study focuses on the observable behaviors and traits exhibited by leaders, we have revised the terminology throughout the manuscript to consistently refer to ‘leadership styles’ rather than ‘approaches.’ This change ensures greater accuracy and uniformity in our presentation.
Comment 7
“On page 3, lines 93 and 94, - “While improvement represents an approach that encourages continuous improvement…” – is unclear. The sentence evidence circularity and lack of specificity because it uses “improvement” to define “improvement” and does not clarify the concept for the reader. What kind of improvement is it referring to?”
Response:
Thank you for your valuable feedback. We acknowledge that the original sentence was vague and exhibited circular reasoning. To clarify the concept, we have revised the sentence to more explicitly define the type of improvement being referred to. The revised sentence can now be found on page 3, lines 106–108.
Comment 8
“On page 3, lines 115 to 119 – The sentence is too long and hard to understand. What “they” meant/refers to is not comprehensible.”
Response:
Thank you for pointing this out. We agree that the original sentence was overly long and that the reference of the pronoun “they” was unclear. To enhance clarity and readability, we have revised the sentence by splitting it into two parts and explicitly referring to “these dimensions” instead of using the pronoun. The revised sentence can now be found on page 3, lines 130–134.
Comment 9
“On page 4, lines 157 to 160 – The sentence is long; breaking or restructuring it will improve readability.”
Response:
Thank you for your helpful suggestion. In response, we have revised the sentence by breaking it into shorter, more manageable sentences to improve clarity and readability. The revised version can be found on page 4, lines 170–175.
Comment 10
“On page 4, lines 167 to 170 - The repetition of "them" and "their" in the sentence (especially so close together) can introduce some mild ambiguity or at least slow down the reader as they mentally track who "them" and "their" refer to.”
Response:
Thank you for your valuable feedback. To improve clarity and avoid the potential ambiguity caused by the repeated use of “them” and “their”, we have revised the sentence. The structure has been changed to use a plural subject, which enhances both readability and precision. The revised sentence can now be found on page 4, lines 178–179.
Comment 11
“On page 7, line 334, when you wrote, “In this respect, leadership styles can be considered as important job resources in the business environment,” consider expanding the idea to the school organization.”
Response:
Thank you for your suggestion. We have revised the sentence to better align with the educational context of the study. The revised version now emphasizes the relevance of leadership styles as important job resources within the school organization. The updated sentence can be found on page 7, lines 346–349.
Comment 12
“On page 14, line 540, the section 8 uses capital letters. Harmonize according to the other sections. Section 8 is more than just conclusions; consider redefining and adopting ‘discussion and conclusions’.”
Response:
Thank you for your comment. We have revised the formatting of Section 8 to align with the style of the other section headings. Additionally, we have updated the title of Section 8 to “Discussion and Conclusions” to more accurately reflect the content.
Comment 13
“On Page 15, line 564, you refer to ‘individuals’. Your study was done with teachers, so I think you should emphasize that. All your findings concern the teaching profession and educational leadership. Apply this suggestion throughout the discussion of results.”
Response:
Thank you for your valuable suggestion. In response, we have revised the discussion section by replacing general terms such as “individuals” and “employees” with the more specific term “teachers.” This revision ensures consistency and emphasizes that our findings are focused on the teaching profession and educational leadership.
Comment 14
“On page 15, line 579, a ‘period’ is missing in the sentence. Consider starting a new paragraph.”
Response:
Thank you for pointing that out. We have corrected the punctuation issue by adding the missing period and have revised the structure by starting a new paragraph with the sentence, as suggested.
Comment 15
We have carefully revised the manuscript to enhance the clarity, flow, and conciseness of the language. The text has been refined to ensure a more direct and effective presentation of our ideas. We appreciate your feedback, which helped us improve the overall quality of the manuscript.
Response:
We have carefully revised the manuscript to enhance the clarity, flow, and conciseness of the language. The text has been refined to ensure a more direct and effective presentation of our ideas. We appreciate your feedback, which helped us improve the overall quality of the manuscript.